# The Impact of Estrogen Receptor in Arterial and Lymphatic Vascular Diseases

**DOI:** 10.3390/ijms21093244

**Published:** 2020-05-04

**Authors:** Coralie Fontaine, Florent Morfoisse, Florence Tatin, Audrey Zamora, Rana Zahreddine, Daniel Henrion, Jean-François Arnal, Françoise Lenfant, Barbara Garmy-Susini

**Affiliations:** 1INSERM-UPS UMR U1048, Institut des Maladies Métaboliques et Cardiovasculaires, Université de Toulouse, BP 84225, 31 432 Toulouse cedex 04, France; coralie.fontaine@inserm.fr (C.F.); florent.morfoisse@inserm.fr (F.M.); florence.tatin@inserm.fr (F.T.); audrey.zamora@inserm.fr (A.Z.); rana.zahreddine@inserm.fr (R.Z.); jean-francois.arnal@inserm.fr (J.-F.A.); francoise.lenfant@inserm.fr (F.L.); 2INSERM U1083 CNRS UMR 6015, CHU, MITOVASC Institute and CARFI Facility, Université d’Angers, 49933 ANGERS Cedex 09, France; daniel.henrion@univ-angers.fr

**Keywords:** estrogens, ERα, endothelial cells, arterial, lymphatic

## Abstract

The lower incidence of cardiovascular diseases in pre-menopausal women compared to men is well-known documented. This protection has been largely attributed to the protective effect of estrogens, which exert many beneficial effects against arterial diseases, including vasodilatation, acceleration of healing in response to arterial injury, arterial collateral growth and atheroprotection. More recently, with the visualization of the lymphatic vessels, the impact of estrogens on lymphedema and lymphatic diseases started to be elucidated. These estrogenic effects are mediated not only by the classic nuclear/genomic actions via the specific estrogen receptor (ER) α and β, but also by rapid extra-nuclear membrane-initiated steroid signaling (MISS). The ERs are expressed by endothelial, lymphatic and smooth muscle cells in the different vessels. In this review, we will summarize the complex vascular effects of estrogens and selective estrogen receptor modulators (SERMs) that have been described using different transgenic mouse models with selective loss of ERα function and numerous animal models of vascular and lymphatic diseases.

## 1. Introduction

Estrogen receptors (ERs) are activated by the steroid female hormone estrogens which regulate reproductive function and maintain numerous tissue homeostasis, in particular in the vessels. There are two main Estrogen Receptors, ERα and ERβ. More recently, G protein-coupled estrogen receptor 1 (GPER) has been proposed as a third ER. GPER has been reported to mediate some beneficial cardiovascular effects of estrogens such as protection against atherosclerosis and hypertension but these results still controversial due to conflicting results obtained with the different mice models targeting this receptor [1,2,3]. Nevertheless, ERα, but not ERβ, mediates most of the vascular effects of estrogens, both on vascular and lymphatic systems particularly in endothelial cells of these vessels [4]. Several studies have demonstrated the presence of functional ERα and ERβ receptors in human and animal vascular cells, including vascular smooth muscle and endothelial cells [5,6,7,8,9]. 

As a member of the nuclear receptor superfamily, ERα was primarily considered as a transcription factor that controls gene expression in response to its ligand [6,10,11]. Thereby, when full agonist 17β-estradiol (E2) binds to its ligand binding domain (LBD), helix 12 packs against helices 3, 5/6, and 11, forming the coactivator-binding groove recognized by the LxxLL motifs of coactivators. The subsequent recruitment of transcriptional coactivators to the activation function AF-1 and/or AF-2 in the ligand-binding domain of ER leads to enzymatic activities involved in chromatin remodeling, histone posttranslational modifications, facilitating ER binding to DNA through DNA binding domain (DBD) and initiating transcription and RNA elongation (Figure 1). Alternatively, ERs can also modulate transcription through a *tethered* mechanism, whereby receptors do not directly bind DNA but instead interact with other transcription factors such as AP-1 or SP1. Besides these nuclear actions, estrogens mediate also rapid signaling through the activation of a pool of ERα localized at the plasma membrane. These membrane-initiated steroid signals (MISS) have particularly been described in endothelial cells (Figure 2). This membrane subpopulation of ERα is localized to endothelial cell caveolae where they are coupled to endothelial nitric oxide synthase (eNOS) in a functional signaling module [12]. Estrogen acutely stimulates eNOS in Human Umbilical Vein Endothelial Cells (HUVEC) or ovine endothelial cells to rapidly stimulate nitric oxide (NO) production [13,14]. 

Recently, ERα, but not ERβ, has been found in lymphatic endothelial cells [15]. Both nuclear and membrane ERα activities were detected to stimulate gene regulation and signaling in the lymphatic endothelium. The lymphatic system is comprised of a network of blind-ended, thin-walled lymphatic capillaries and collecting vessel, which drain fluids to maintain tissue homeostasis. The main function of the lymphatic vasculature is to return fat, fluid, macromolecules and cells, such as extravasated leukocytes, back to the blood circulation through the subclavarian vein. The collected lymph is filtrated through the lymph nodes that provide an interface for antigen-presenting cells to process antigen and to initiate immune response.

Lymphatic capillaries are present in all the vascularized tissues, with the exception of the central nervous system, bone marrow and retina. Lymphangiogenesis, the growth of new lymphatic vessels, is crucially involved in the pathological conditions such as heart ischemia, tumor metastasis, and chronic inflammatory diseases and a dysfunction of the lymphatic function leads to the development of a severe condition named lymphedema [16,17]. Recently, it has been shown that inhibiting ERα in the lymphatic endothelium is a severe risk factor for secondary lymphedema.

Based on the unique pharmacology of the ERs, selective ER modulators (SERMs) have been developed to stimulate or inhibit estrogen-like action according the tissue considered. These drugs act as agonists or antagonists in the different cell type due to characteristic alterations in the conformation of the ligand-binding domain of ERα and their ability to interact with co-regulators.

The prototypical SERM is tamoxifen, developed in the 1970s for breast cancer risk reduction in women [18,19]. Indeed, tamoxifen induces a peculiar orientation of helix 12 that prevents the conformation allowing to expose a coactivator-binding groove and coactivator recruitment thanks to this activation function AF-2. Furthermore, this distinctive structure of AF-2 strengthens the recruitment of corepressors and thereby of the repressive machineries. However, at the same time, Tamoxifen also exerts agonist effects that mimics estrogen action in others tissues such as uterus, bone and arteries. *In vitro* experiments demonstrate that this agonist activity is entirely mediated by the other activation function (AF) named ERα AF-1. The crucial respective roles of ERα-AF1 and ERα-AF2 in vascular protection *in vivo* were directly demonstrated using ERα-AF1^0^ mice and ERα-AF2^0^ [20,21,22]. More recently, decoupling of nuclear and membrane ERα activity has been proposed as a new possible option to optimize ERα modulation through the development of pharmacological tools that specifically activate membrane, but not nuclear, ERα: Estrogen-dendrimer conjugate (EDC) [23] and “pathway preferential estrogens” (PaPEs) [24] efficiently activate membrane-initiated steroid signaling (MISS) but do not induce nuclear ER transcriptional activity. By contrast, we demonstrated that the fetal estrogen estetrol (E4) acts as a natural modulator of ERα through a selective activation of nuclear ERα antagonizing ERαMISS effects [25].

According to the importance of ERα in vascular diseases and in women health and medicine (contraception, hormonal treatment of menopause, breast cancer and osteoporosis treatments, etc.), we propose here to summarize our current understanding of vascular action mediated by ERα and the progress that have been done using new mouse models. This review will focus on the impact of estrogen receptors on arterial and lymphatic vasculature in pathological conditions.

## 2. Importance of Estrogens in Sex Difference

A lot of epidemiological studies evidence that men and women develop differently cardiovascular diseases (CVD). Men develop CVD at younger age, with more severe coronary artery diseases, while women develop CVD later in life, with different outcomes [26]. However, hormonal replacement therapy was initially shown to increase risk of coronary diseases in the prospective analysis from the Women’s Health Initiative observational (WHI) study [27]. The increased number of coronary events in this WHI study, which enrolled more than 16,000 postmenopausal women aged 50–79 years (at time of study enrollment) over 15 years, was finally attributed to the delay between menopause and the initiation of the treatment while the hormonal treatment was found to be protective when postmenopausal women are treated in the years following onset of menopause [28]. Sex hormones might not be the only determinant in sex differences in CVD. Recently, Alsiraj Y. and his collaborators demonstrate that having an XX chromosome augments atherosclerosis in both male and female mice and promotes dyslipidemias [29]. This finding could contribute to explain the higher prevalence of atherosclerosis in aged, postmenopausal females and underline the importance of sex chromosomes on the cause of sex differences in the development of cardiovascular diseases [30]. Moreover, epigenetics mechanisms control sex-specific gene expression in these cardiovascular diseases and the impact of epigenetic on ischemic stroke, myocardial infarction, coronary heart disease and coronary artery disease has been specifically reviewed and remained an open debate [31]. The role of androgens, and in particular of testosterone, also greatly affects these cardiovascular diseases, but will not discussed here [32]. Clinical and preclinical studies also provide considerable evidence that estrogens have beneficial effects on risk factors for CVD such as circulating lipid profiles (i.e., increase of HDL), metabolic regulations and diabetes [33], while these effects are lost in post-menopausal women [34]. The use and the effects of alcohol consumption and cigarette smoking, two other major risk factors of CVD, also differ significantly between men and women [35]. Regarding hypertension, a major risk factor that often precedes more serious CVD, there is also an increased prevalence in men and post-menopausal women as compared to premenopausal women, suggesting a role of ovarian hormones in blood pressure regulation [36]. The prevalence of hypertension is also greater in women with ovarian deficiency than in age-matched premenopausal women. Animal models also demonstrated that blood pressure is increased in ovariectomized animals or in mice deficient for ERα [37]. All these studies suggest that estrogens are responsible for the lower incidence of hypertension in women before menopause. Arterial stiffness is often associated with hypertension, but also with risk of other cardiovascular diseases, such as lipid disorders and diabetes. Interest in this arterial stiffening has increased dramatically in the past decade, due to the possibility of using measures of wave reflection for approaching the arterial stiffness [38]. Recently, Ute *et al.* (2020) suggested important sex differences in these measures of arterial wave reflection, pointing endogeneous estradiol levels as potential actors that could increase vasotonus of the small and medium arteries [39]. 

## 3. Estrogen Receptor (ERα) and Blood Endothelial Healing in Response to Vascular Injury

A large body of evidence indicates that estrogens induce vascular repair following injury to large arteries, such as the carotid or coronary arteries during balloon angioplasty or stents. Estradiol was shown to increase both migration and proliferation of endothelial cells *in vitro* [4,40,41]. These E2 actions on migration and proliferation of endothelial cells are an essential aspect of vascular healing that has been studied extensively in experimental animal models. First, in ovariectomized female rats, E2 was first reported to accelerate endothelial regrowth after balloon denudation of the carotid artery [42]. Using the model of electric injury of mouse carotid, it was demonstrated that this beneficial endothelial effect of estrogens was mediated by ERα, but not by ERβ [43]. This process is accelerated by E2 as a consequence of the retrograde commitment of an uninjured endothelial zone to migrate and proliferate, contributing to an enlargement of the regenerative area [44]. These models were also used to demonstrate that E2 induces a large program of regeneration, involving growth factor FGF-2 (fibroblast growth factor 2) [45].

E2 also stimulates the endothelial NO synthase (eNOS) to induce production of estrogen nitric oxide (NO), a radical messenger that elicits an important vasculo-protective role through its vasodilating and anti-aggregating properties. Induction of NO production by acute estrogens was first observed on the rabbit femoral artery, where E2 increases acetylcholine-induced vasorelaxation. In addition, chronic E2 can increase NO bioavailability through reduced generation of reactive species and decreased breakdown of NO [46,47]. This E2 stimulation of eNOS is rapid and involves non genomic signaling such as an increase of intracellular Ca2+, as shown in fetal pulmonary arterial endothelial cells [48]. Activation of the tyrosine MAP kinase signaling has been implicated in the mechanism of ER-mediated eNOS activation [14], while others reported the role of the Pi3-kinase-Akt pathway and demonstrated interaction between ERα and p85 subunit of PI3kinase [49]. Finally, more recent studies have also pointed interaction of the plasma membrane ERα with Gαi proteins due to interaction with the specific amino acid sequence 251–260 [50,51,52]. Additionally, the striatin directly targets ERα to the cell membrane after binding to amino acids 183–253 in order to act as a scaffold for the formation of an ERα-Gαi complex [53]. ERα-Gαi interaction also leads to the regulation of endothelial hydrogen sulfide (H_2_S), known to induce vasorelaxation, to stimulate endothelial cell-related angiogenic properties and to protect against atherosclerosis [54]. Again, this E2-dependent increase in NO production by the endothelium in mice endothelium is entirely mediated by ERα, but not by ERβ [14,55], as demonstrated in a model of total ERα disruption [56]. This result is in contrast with those obtained using the initial model of ERα knock-out (ERαKO by insertion of the Neomycin-resistance gene cassette in exon 1 in which the E2 effect on NO production was preserved) [57] (Table 1). The persistence of the vascular protection in the latter ERαKONeo model was attributed to a non-natural mRNA alternative splicing that results in the expression of a truncated chimeric isoform of 55 kDa, deficient in ERα-AF1 box 2 and 3 [58,59]. This E2-induced production of NO is fully inhibited by the ER antagonists, such as tamoxifen and ICI-182,780 [48]. The accelerative effect of E2 following both perivascular and endovascular injury is lost in eNOS knock-out mice [60]. However, inhibition of NO synthase by *N*-Nitro-l-arginine Methyl Ester (L-NAME) does not block endothelial healing, showing that the presence of eNOS protein, but not eNOS activity, is critical in the endothelial cells of arteries [60]. Tamoxifen fails to accelerate reendothelialization after perivascular carotid electric injury [61], but its action on subsequent endovascular injury has not yet been reported.

The use of mouse models with cell-specific loss of function of ERα (using the CRE:Lox strategy) indicates that ERα expression is required in both endothelium and hematopoietic cells for the accelerative effect of E2 on reendothelialization [62]. However, which cell type and/or circulating factors are required from the hematopoietic compartment remains unknown. The estrogen–dendrimer conjugate (EDC) is excluded from the nucleus and activates only non-nuclear signaling. Chambliss and collaborators [63] demonstrated that activating only the membrane actions of ERα is sufficient to confer a cardiovascular protection: both E2 and EDC stimulates endothelial cell proliferation and migration in an ERα and G protein-dependent manner, accelerated reendothelialization after electric injury, and attenuated the development of neointimal hyperplasia following endothelial injury in a context of hypercholesterolemia (Table 1). These E2- and EDC- induced accelerative effects on re-endothelialization were abolished in ERα-C451A mice while both E2 and EDC effects on acceleration of endothelial healing were fully preserved in the nuclear loss of function mice (ERα-AF2^0^), confirming that the MISS effects were necessary but sufficient to accelerate endothelial regeneration after electric injury [21,64]. More recently, molecules which preferentially activate extranuclear-initiated signaling and not nuclear/genomic effects of ERα were designed, called “pathway preferential estrogens” (PaPEs). PaPEs have a similar effect than E2 on acceleration of endothelial healing, and further confirmed the membrane actions of E2 for both the rapid NO production and acceleration of endothelial healing following perivascular injury, suggesting that activation of membrane signaling can promote vascular protection without increasing cancer risk [24,63]. 

Finally, other studies demonstrate that smooth muscle cells in the vessels are also targeted by estrogens, preventing the smooth muscle cells proliferation, resulting in the reduction of the medial hyperplasia in a model of wire carotid injury [5,65]. This medial hyperplasia is again ERα dependent [66] and requires extra-nuclear signaling as recently shown by Bernelot et al., who used a mouse model in which rapid signaling is blocked by overexpression of a peptide that prevents interaction of ERα with the scaffold protein striatin [67]. Disruption of ERα/striatin interactions abrogates the E2 protective effect of medial hyperplasia. Additionally, the same wire carotid injury performed in hypercholesterolemic ApoE deficient mice indicated that selective activation of extranuclear actions of ERα by EDC inhibits smooth muscle cell (SMC) hyperplasia, now not in the media but in the intima [63]. Furthermore, the wire injury performed not at the carotid, but at the femoral artery, demonstrated again the inhibitory effect of estrogens on neointimal SMC proliferation, reducing the neointimal hyperplasia observed in ovariectomized mice [22]. However, combining the use of transgenic mice and pharmacological tools, the preventive action of E2 on neointimal hyperplasia was found to be mediated by SMC through ERα-AF1 [22]. Taken together, these data indicate that proliferation of neointimal SMC in the femoral muscular artery is controlled by ERα-AF1 nuclear actions, while the SMC medial hyperplasia in the elastic carotid artery is mediated by ERα rapid actions. The differences in these two models are likely attributable to the difference in the structure between femoral and carotid arteries, SMC phenotype, and the complexity of the SMC signaling, but also potentially of EC-SMC interactions in these different localizations, demonstrating the difficulty of modelizing the vascular physiopathology. 

## 4. ERα and Arterial Remodeling and Response to Ischemia

Endothelial cells sensing shear stress and a chronic increase in blood flow in resistance arteries induce outward remodeling (increased lumen diameter) in order to normalize the shear stress. This remodeling is also associated with increased wall thickness, aiming at normalizing the tensile stress induced by the diameter expansion [80,81]. In addition to angiogenesis and arteriogenesis, this high flow-mediated remodeling of resistance arteries plays a key role in revascularization of ischemic tissues after occlusion of a large artery [82]. This remodeling requires a transient and moderate inflammatory [83], and oxidative response [84] followed by the activation of the NO pathway [85]. A chronic increase in blood flow occurs in physiological situations, such as tissue growth, pregnancy, or sustained exercise, and it is required for collateral artery growth. On the other hand, flow-mediated outward remodeling is severely reduced in diabetes [86], hypertension [87] and aging [87,88]. There is also an important sex difference in remodeling. Indeed, flow-mediated remodeling is absent in 12-month-old male rats [89], whereas it is present in 12-month-old female rats [90], although the amplitude of the remodeling is proportional to the circulating level of E2, and thus decreases progressively with age [91]. In addition, it is absent in young ovariectomized female rats and in young mice lacking ERα [68]. Female sex hormones also have an important role in uteroplacental adaptation during pregnancy with an important uterine artery outward remodeling involving trophoblast invasion, hyperplasia and hypertrophy, and changes in extracellular matrix composition [92]. E2 also stimulates endometrial secretion of a factor(s) that promotes vascular smooth muscle cell migration as an early step in blood vessel outward remodeling within the endometrium [93].

The essential role of E2 and endothelial ERα on arterial flow-mediated outward remodeling was demonstrated using mice with Tie2- promoter to express the Cre-recombinase [68]. Furthermore, this remodeling is controlled by the nuclear activating function 2 of ERα. Indeed, ERα-AF2^0^ and C451A-ERα mice with selective inactivation of nuclear and membrane ERα, respectively, made it possible to demonstrate that nuclear ERα plays a prominent role in the promotion of arterial remodeling, as well as in the prevention of angiotensin II-dependent hypertension [72]. This was further confirmed using specific selective activation of nuclear ERα with estetrol (E4), a fetal estrogen that prevents AngII-induced hypertension and favors flow-mediated remodeling [72].

This involvement of E2 in flow-mediated outward arterial remodeling concurs to the protective actions of E2 against ischemia and the damage of ischemia/reperfusion injuries that can concern many tissues such as the myocardium, hindlimb, brain and skin [94,95,96]. E2-mediated protection was also found to decrease vascular leakage and reperfusion hemorrhages, acting both through an activation of arteriogenesis and a reduction of apoptosis [95]. The anti-apoptotic effects of E2 have largely been demonstrated *in vitro* on endothelial cells [97], and *in vivo* against brain ischemia [98] or myocardium ischemic/reperfusion injury, where endothelial ERα plays an essential role in the coronary and myocardial protective effects of E2 in ischemia/reperfusion [99]. Finally, non-nuclear ERα activation was recently shown to reduce cardiac ischemic reperfusion injury [100] and stroke severity in mice [77].

Whereas outward remodeling of arterioles participates in the revascularization of ischemic tissues, ERα also exerts a protective effect on large arteries. Indeed, the vascular structure was investigated in female mice lacking specifically ERα in endothelial cells (VE-Cad promoter driven expression of Cre-recombinase) and fed a Western diet (WD) [74]. Arterial stiffening was evaluated *in vivo* using pulse wave velocity Doppler ultrasound. Interestingly, arterial stiffening induced by the WD was absent in both male and female mice lacking endothelial ERα [74,75]. Absence of ERα in endothelial cells in WD-fed mice was also associated with hypertrophic remodeling in mesenteric resistance arteries. These experiments further show that E2, through endothelial ERα activation, exerts a protective role favoring outward remodeling of resistance arteries in ischemic diseases and preventing arterial stiffening in WD-fed mice. 

## 5. ERα and Atherosclerosis

The atheroprotective action of estrogens has been demonstrated in numerous animal models of early atheroma. In particular, E2 has been shown to strongly prevent arterial lipid deposition in several animal models including monkeys, rabbits, and atheroprone (apolipoprotein E–deficient [ApoE^−/−^] and low-density lipoprotein receptor-deficient [LDLR^−/−^]) mice [73,101]. Different groups have demonstrated the crucial role of ERα in mediating this atheroprotective effect of estrogens [102,103]. Since selective activation of ERαMISS using EDC increases NO production and accelerates endothelial healing after carotid injury without impact on sex targets, Shaul’s group initially proposed that decoupling membrane and nuclear ER activation could provide cardiovascular protection without increasing cancer risk [63]. However, the use of mouse models in which nuclear effects are abrogated (ERα-AF2^0^/LDLR^−/−^ mice) highlights the requirement of nuclear/transcriptional actions of ERα for early atheroprotection at the level of the aortic sinus [21], but also for the prevention by endogenous estrogens through en face analysis of the thoracic and abdominal aorta [72]. By contrast, ERαC451A mice were fully responsive to estrogens to prevent atheroma, whereas selective activation of ERαMISS with EDC [63] or PaPE [72] does not confer atheroprotection. Thus, the increase in endothelium-derived NO and the atheroprotective effect of estrogens involve different activation pathways of ERα (ERαMISS *versus* nuclear ERα activation respectively). Interestingly, this is quite consistent with the absence of alteration of the E2 atheroprotective effect by NOS inhibition [104] and in hypercholesterolemic eNOS^−/−^ mice [105,106] reported twenty years ago.

Nevertheless, several lines of evidence identify endothelial cells as a major target for estrogens. Indeed, destruction of the endothelial layer in the aorta using a balloon led to complete loss of anti-atherogenic effects of estrogens in hypercholesterolemic rabbits. The central role of endothelium was then fully demonstrated targeting ERα on endothelial cells using the CRE/Lox system under control of the Tie promoter [73,101]. *In vitro*, estrogens have been described to exert beneficial effects on endothelial cells that could participate to their atheroprotective actions such as inhibition of senescence [107] and apoptosis [97,108], suppression of vascular cell adhesion molecules [109] and cytokine expression [110], as well as antioxidant effects [111]. In addition, the tight junction Claudin-5 was identified as an estrogen target in the endothelium both *in vivo* and *in vitro*. This increase in endothelial junctional protein levels may improve vascular integrity and barrier function against oxidized Low density lipoprotein (oxLDL) infiltration. More recently, growing evidence has established regulation of numerous microRNA (miRNA) involved in the regulation of endothelial function by E2 through ER, providing new insights by which estrogen can modulate vascular action [112]. This study validated the presence of estrogen response element (ERE) in the regulatory region of endothelial miRNA, showing E2-induced increase of miR-30b-5p, miR-487a-5p, miR-4710, miR-501-3p and decreased of miR-378h and miR-1244. miRNA may therefore act as crucial epigenetic regulators of gene expression induced by E2 in endothelial cells and could contribute to further characterize the role of specific miRNA in vascular actions mediated by E2 in physiological and pathological conditions [113].

Since development of atherosclerotic lesions involves complex interplay, numerous other endothelial mechanisms could participate to E2-induced atheroprotection. In particular, ApoE^−/−^ Rag^−/−^ double-deficient mice were no longer protected against atherosclerosis by E2 [114]. Estrogens act also on macrophages and ERα within these cells modulates inflammatory response, maintains oxidative metabolism and regulates reverse cholesterol transport [115]. These findings underline the complex interactions that link endothelial cells and hematopoietic-derived cells during the atherosclerosis process. 

### Selective Estrogen Receptor Modulators (SERMs) and Atheroprotection

Clinical data suggest that the prototypical SERM tamoxifen, used in the hormonal treatment of breast cancer, could slow the progression of atherosclerosis in postmenopausal women as indicated by changes in carotid intima media thickness [116]. The atheroprotective action of tamoxifen was reported in several experimental models in monkeys [117], rabbits as well as in ApoE^−/−^ [118] and LDLR^−/−^ [61] mice. In mice, this effect is mediated by ERα but, whereas ERαAF-1 is dispensable for the atheroprotective action of E2, ERαAF-1 is absolutely required to induce atheroprotection in response to tamoxifen [61]. Thus, even if the atheroprotective effect of TAM and E2 are both mediated by ERα, the subfunctions of the receptor accounting for this beneficial vascular action are different. Cellular(s) target(s) and molecular pathways accounting for the respective effects of these two molecules remain to be determined. Finally, better understanding of the mechanism underlying tamoxifen vascular agonist action is important since a recent scientific statement from the American Heart Association on CVD highlighted the closest connection between CVD and breast cancer.

## 6. ERα and Lymphatic Vasculature

So far, the lymphatic system has been poorly studied because of the difficulty of differentiating its cells from the arterial and venous cells. However, the identification of specific markers such as Prox-1, Lyve1 and Podoplanin twenty five years ago allowed the investigation of the specific features of lymph vessels [119,120]. 

Whereas it is now admitted that there was a significant correlation between the decrease in lymphatic pumping and age, the relation to sex is less established. In females, it is more difficult to attribute the defect in lymph pumping to age or to hormonal status because of menopause that generates strong hormonal changes in middle-age women. Some randomized studies indicate that the age-related decrease in lymphatic pumping pressure is more pronounced in females of postmenopausal age [121]. However, when surgical procedures such as lymphaticovenous anastomosis are performed in lymphorrhea patients, no significant differences in the lymphatic function are found in patient age or sex [122]. Recent findings would underscore the hypothesis that gender difference between male and female could be due to the basal phenotype of the lymphatic system as the lymphatic density is higher in female than in male in several organs such as the heart [123,124]. In particular, female mice have more cardiac lymphatics in the basal state than males [123]. This could explain a distinct sex-dependent difference in the development of myocardial edema as female exhibit a short-term cardioprotection induced by a delay edema response. It is interesting to note that a complete recovery observed in both male and female independently of the basal phenotype of increased cardiac lymphatics suggesting that the lymphatic gender difference might be more important in the ethology of the pathology [124]. Nevertheless, this difference cannot be attributed to a difference in ERα gene expression as similar patterns are found in male and female in heart, aorta, liver, diaphragm, lymph nodes, skin and lung [15]. The sex difference responses to vascular injury can be also attributed to different expression of lymphangiogenic factors depending on metabolic status. In particular, serum vascular endothelial growth factor (VEGF)-D shows sexual dimorphism with higher circulating levels in lean females than in lean males. On the contrary, obese females have higher VEGF-C levels than obese males. These factors may be controlled by metabolic status independently of the estrogen levels and could modify the lymphatic physiologic response, especially with regard to immune response [125].

As observed in the endothelium of the blood vessels, ERα is expressed on the lymphatic endothelium, which suggests a control of the lymphatic function mediated by estrogens [15]. The effect of E2 in promoting lymphatic marker genes expression such as Prox1 and Flt-4 was first described in a human lymphatic endothelial cell line expressing the large T antigen of simian virus 40 [126], but the exact localization of the estrogen responsive element (ERE) on the promoter of these genes was found recently by Morfoisse and collaborators [15]. 

Despite a substantial amount of evidence suggesting control of the lymphatic endothelium function by estrogens, the lack of genetically modified experimental models has considerably slowed the understanding of subjacent molecular mechanisms. During the estrous cycle, the role of lymphatic vessels located in the uterine adventitia was restricted to the transport of estrogen to the periarterial nerves to participate to the control of vasodilatation [127]. The paracrine effect of lymphatic vessels in the endometrium was also attributed to their close association with spiral arterioles without evidence of a direct control by the estrogen of the lymphatic endothelium [128].

Although endocrine organs including adrenal gland and ovary are responsible for *de novo* production of estrogens, it can also be synthetized in numerous organs including the brain, placenta, and adipose tissue. Whereas the role of the lymphatic system in adrenal gland seems to be limited to the systemic transport of hormones, its function can be directly modulated by estrogens in the ovaries [129]. Estrogens promote the opening of the lymphatic vasculature into the stroma of the ovarian bursa to improve the lymphatic drainage and maintain the ovarian microenvironment suitable for ovulation [129]. Altogether, there is a large amount of evidence suggesting the control of the lymphatic function by ER. However, less is known about its entailment during the lymphatic dysfunction even though particular lymphatic vascular malformations, which frequently occur during puberty, suggest a role of hormones and hormone receptors in the etiology of the pathology.

## 7. ERα and Lymphedema

When the lymphatic system becomes dysfunctional, it leads to lymphedema, an impaired lymphatic return and swelling of the extremities by accumulation of fat and fibrosis in the arm or in the leg [15,130]. It can be an inherited condition (primary lymphedema) or can occur after cancer surgery and lymphadenectomy or filarial infection (secondary lymphedema). Hereditary lymphedema have been categorized in two groups depending of their age of onset: congenital lymphedema develop at birth (Milroy disease) or lymphedema praecox, which occur around puberty (Meige disease, distichiasis syndrome) [131,132]. Whereas the influence of estrogen is thought to be an important etiological factor in primary lymphedema developing at puberty, little is known about the role of estrogen receptors in lymphatic dysfunction [133]. It has now been clearly established that primary lymphedema is sex-linked with an average ratio of one male to three females, but the role of hormones, in particular estrogens, remains elusive [130,133]. A large proportion of primary lymphedema such as Turner syndrome are directly linked to a defect in female hormone synthesis, as this affects ovarian development [134]. It is a wide consensus to provide estrogen supplementation to improve health care delivery for these patients [135]. Although lymphedema-distichiasis occurs at birth, onset of lymphedema is delayed until puberty, suggesting a role of hormonal balance in the induction of the pathology [136]. Other rare lymphatic diseases are also associated with estrogen. Pulmonary lymphangiomyomatosis is characterized by a strong proliferation of smooth muscle around the lymphatic vessels of the lung, mediastinum, and retroperitoneum. It occurs only in women of child-bearing age and is manifested by spontaneous pneumothorax and chylous pleural or abdominal effusion [137,138]. Lymphangiomyomatosis indicates the involvement of hormones–estrogens and suggests the use of ER inhibitor therapy to stop the progressive course of the pathology and to control the chylothorax.

In western countries, most secondary lymphedema are induced by cancer treatments, in particular, fifteen to twenty percent of breast cancer survivors develop lymphedema months, and sometimes years, after surgery, suggesting that lymphedema is not only a side effect of the surgery, but involves the effect of cancer treatment such as chemo-, radio-, and hormone therapies [139,140,141]. A recent study demonstrated the role of ERα in secondary lymphedema (Table 2) [15]. The estrogens limit the lymphatic dermal back flow induced by the leakage of the large collecting vessels. Additionally, the endothelial knock-down of ERα promotes an increased swelling in the limb in a mouse model of lymphedema and is associated with hyper-dilated overloaded capillaries. This study demonstrated that ERα promotes the transcription of lymphangiogenic genes in lymphatic endothelial cells (LECs) including VEGFR-3, VEGF-D and Lyve-1 that possess ERE and AP1 domain in the promoter region. The upregulation of lymphangiogenic genes induced by E2 was observed both *in vitro* and *in vivo* leading to a stimulation of lymphangiogenesis [15]. Importantly, as observed in blood endothelial cells, E2 stimulates the non-genomic effect of ERα, as shown by an increase of Erk signaling in LEC. Altogether, this study strongly suggests that blocking ERα affects lymphatic shape and function. 

### SERMs and Lymphatic Diseases

It has been established for decades that estrogen receptor expression increases the risk of lymphatic invasion in ovarian and breast cancer. This was attributed to the production of lymphangiogenic growth factors by the tumor cells to promote the development of the peritumoral lymphatic vasculature. However, the role of ER on the lymphatic endothelium has been poorly investigated. Up to eighty percent of breast cancers are ER-positive and have to undergo a hormone therapy, and it is tempting to speculate on a role of hormone therapy on the lymphatic tissue [142]. Tamoxifen, a partial agonist of ER, has been the most commonly used hormone therapy for decades for premenopausal women with breast cancer [18], whereas after menopause, women switch to an aromatase inhibitor that blocks the conversion of testosterone into estrogens. Based on recent evidence showing the presence of ERα in lymphatic endothelial cells, it is reasonable to postulate that SERMs and/or Aromatase Inhibitors could have an impact on lymphatic-related breast cancer side effects such as lymphedema [15]. Despite a first function aiming to target epithelial cells, the effect of tamoxifen on the endothelium has been explored in coronary artery disease prevalence [143]. Additionally, recent evidence is converging towards a role of hormone therapy on the lymphatic endothelium [15]. However, research into the role of hormone therapy in lymphedema remains limited. Recently, a study aiming to investigate the treatment-related risk factors for lymphedema in women who had breast cancer revealed the first evidence of a correlation between hormone therapy and lymphedema [144]. This study associates long-term tamoxifen treatment with arm lymphedema in obese women and suggests that lymphedema risk may be an indication to consider a weight reduction in breast cancer survivors. Lymphedema in the legs has also been attributed to the use of tamoxifen which may contribute to the development of deep vein thrombosis [145]. In an experimental mice model of lymphedema, tamoxifen induced lymphatic capillaries leakage and aggravates lymphedema [15]. By blocking the expression of ER-targeted genes in LEC, tamoxifen exhibits both a direct and indirect effect on the lymphatic endothelium by targeting VEGFR-3/VEGF-D/Lyve-1 and Hyaluronan synthases, respectively [15]. *In vitro*, tamoxifen blocked lymphatic endothelial cells migration and branch-point formation, but had no effect on LEC survival. It also antagonized the membrane non-genomic ERα by blocking the Erk signaling pathway [15]. This study provided evidence for the adverse effect of hormone therapy on the lymphatic endothelium and suggested that it could contribute to the worsening of the lymphedematous condition. Taken together, there is a large body of evidence suggesting that female hormones could modulate lymphatic drainage, and indicating that the hormonal status of patients is critical for determining suitable adjuvant treatment to prevent lymphatic dysfunction.

## 8. Discussion and Future Perspectives

This review summarized some of the various effects of estrogens on vascular and lymphatic vessels, highlighting their protective role in vascular diseases. Indeed, similar to the effect previously observed on the blood endothelium, the protective effect of estrogens could be extended to the lymphatic endothelium. Mice models and pharmacological tools initially identified the extra-nuclear membrane actions of ERα as potential targets for mediating arterial protection using selective membrane activation, such as EDC and PaPEs, while preventing the risk of breast cancer. More recent data have demonstrated that this selective membrane activation is limited to the rapid NO-dependent dilation and to the acceleration of endothelial healing, but is not sufficient to mediate atheroprotection, arterial outward remodeling or collateral growth, protection against angII-induced hypertension, prevention of neo-intimal hyperplasia. Indeed, all these protective effects require the nuclear effects of ERα. Moreover, this review highlights the importance of ERα in protecting both small and large arteries through activation of outward remodeling of résistance arteries allowing revascularization in ischemic disorders and by opposing large arteries stiffening in metabolic disorders. In addition, several studies have evaluated the therapeutic potential of stimulating lymphangiogenesis in different vascular pathologies, including atherosclerosis, hind-limb ischemia and myocardial infarction [146,147,148,149]. While the lymphatic vasculature was first described by Hippocrates (460–377 B.C.), it remains the omitted part of the circulation. Recent studies have demonstrated the role of the lymphatic system in cardiovascular diseases including atherosclerosis, hypertensions, heart ischemia and reverse cholesterol transport. Regarding the central role of lymphatics in the maintenance of fluid homeostasis, it is not surprising to observe sex-related differences associated with lymphatic function. 

The treatment with recombinant VEGF-C/D or adenovirus-mediated overexpression of VEGF-C/D showed promising results in animal models suggesting that a modulation of lymphangiogenesis could improve the cardiovascular function. In that context, it is tempting to speculate that ERα may represent an encouraging target for the lymphatic endothelium-associated diseases. However, the knowledge gained so far about regulating mechanisms of lymphangiogenesis in vascular diseases is largely from animal studies (mainly rodents), which presents the challenge of translating the obtained results to humans. Furthermore, the induction of lymphangiogenesis promotes tumor metastasis, and therefore the safety of lymphangiogenic therapy should be carefully tested in the context of each pathological condition. Considering the beneficial role of estrogens on lymphatic vessels, an estrogen-based treatment might be investigated for lymphedema following ER negative breast cancer. Moreover, current studies are being carried out on another estrogen-family member, fetal estrogen estetrol (E4), which was described in one study to have anti-proliferative effect in patients with advanced ER^+^ breast cancer, and which could be tested with respect to its efficacy in preventing lymphedema after breast cancer. Further mechanistic studies are needed to elucidate the cellular and molecular pathways through which estrogen mediates its vascular protective actions and animal-to-human translational studies should be considered to reveal novel strategies for estrogen rescue. Taken together, this review helps to determine the role of E2 is vascular diseases prevention.

## Figures and Tables

**Figure 1 ijms-21-03244-f001:**
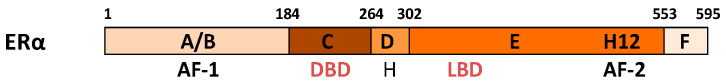
Schematic representation of human estrogen receptors (ERα) protein. ERα protein exhibits six domains, A to F, oriented from the amino (N) to carboxyl (C) terminus. The domains which locate key functions are indicated: activation function (AF)-1 and AF-2 mediate transcriptional activity. The DNA binding domain (DBD) interacts with estrogen responsive element (ERE) DNA motifs, and the ligand binding domain (LBD) binds E2. H is the hinge domain. Helix 12 (H12) interacts with transcriptional activators and repressors following ligand binding.

**Figure 2 ijms-21-03244-f002:**
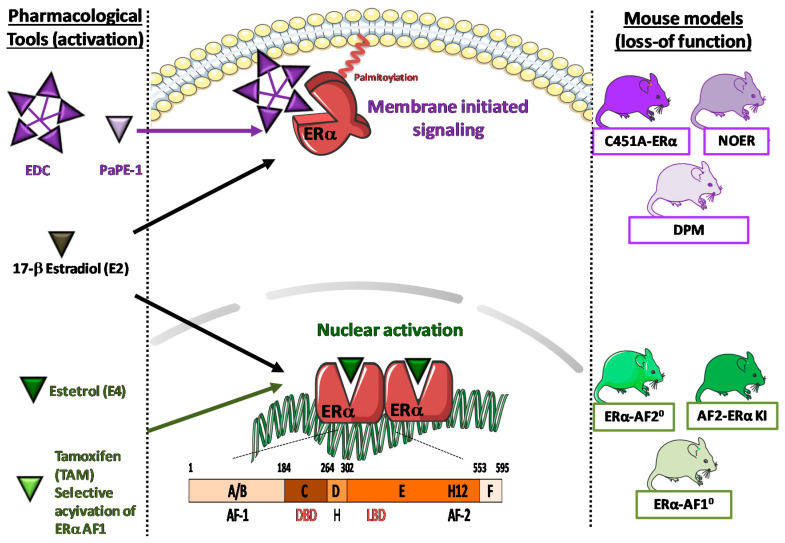
Nuclear/genomic effects of ERα. Nuclear/genomic and rapid extra-nuclear membrane actions (MISS) of ERα are presented in the middle panel, with loss of function mouse models (on the **right**) and estrogenic compounds (on the **left**) activating both nuclear and membrane ERα (E2), selective activation of either extra-nuclear actions of ERα (EDC, Estrogen-Dendrimer Conjugate andPaPE, Pathway Preferential Estrogens) or nuclear actions of ERα (E4, estetrol) or the Selective Estrogen Receptor Modulator (SERM, such as tamoxifen).

**Table 1 ijms-21-03244-t001:** Summary of vascular effects in mouse models of E2 and pharmacological ligands.

Mouse Models	Names	Description	Effect on Vascular Protection	Reference
ERβ- null	ERβ-KO	Global deletion of ERβ by deleting Exon2	Preserved E2-induced acceleration of re-endothelialization, Preservation of arterial flow-mediated remodeling	[43,56,68]
ERα- null	ERα-KO	Global deletion of ERα by deleting Exon2	Loss of endothelial NO production, Loss of E2-induced acceleration of re-endothelialization, Absence of E2-induced atheroprotection, Absence of arterial flow-mediated remodeling, Absence of angiotensin II-induced hypertension, Absence of E2-induced prevention of lymphedema	[15,43,56,59,68,69]
ERα-Neo KO	ERα-KO	Insertion Neo Cassette in exon 1	Preserved endothelial NO production	[59]
ERα- AF1 mutant	ERα-AF1^0^	Deletion of -1-148)aa AF1 domain	Preserved endothelial NO production, Preserved E2-induced acceleration of re-endothelialization following perivascular electric injury, Preserved E2-induced atheroprotection, Abrogation of the neointimal hyperplasia protection induced by E2	[20,22]
ERα- AF2 mutant (loss of nuclear signaling)	ERα-AF2^0^	Deletion of AF2 domain	Loss of endothelial NO production, Loss of E2-induced acceleration of re-endothelialization following perivascular electric injury, Absence of E2-induced atheroprotection, Absence of angiotensin II-induced hypertension	[21,69]
ERα- AF2 mutant	AF2ERKI	Mutation of L L543A, L544A in helix 12	Not determined	[70]
Loss of membrane signaling	C451A-ERα, NOER	Mutation of palmitoylation site (C451A)	Loss of endothelial NO production, Loss of E2-induced acceleration of re-endothelialization following perivascular electric injury, Preserved E2-induced atheroprotection, angiotensin II-induced hypertension and flow-mediated arteriolar remodeling	[64,69,71,72]
Inactivation of extra-nuclear signaling	DPM mice	Overexpression of a peptide preventing striatin interaction	Loss of E2-induced protection against medial hyperplasia following wire carotid injury	[67]
Inactivation of ERα on hematopoietic and lymphatic/ endothelial cells	Tie2-CRE- ERαL2L2	Expression of Cre recombinase under Tie2-promoter	Loss of E2-induced acceleration of re-endothelialization, Absence of E2-induced atheroprotection, Absence of arterial flow-mediated remodeling on a model of ligation of 2 mesenteric arteries, Preserved protective E2 effect against neointimal hyperplasia after wire femoral injury, Absence of E2-induced prevention of lymphedema, Lymphatic leakage and hyperdilated lymphatic capillaries	[15,22,62,68,73]
Inactivation of ERα on endothelial cells	VE-Cad-CRE- ERαL2L2	Expression of Cre recombinase under VE-Cad promoter	Decrease of vascular thickness in WD-fed females and males fed evaluated by pulse wave velocity Doppler ultrasound, Preserved mesenteric remodeling in WD-fed females, Lymphatic leakage and dilated lymphatic capillaries	[74,75]
Selective activation of Membrane ERα	EDC	(estrogen-dendrimer conjugate)	Activation of endothelial NO production, Acceleration of re-endothelialization following perivascular electric injury, Protection on medial hyperplasia after wire carotid injury, No protection against neointimal hyperplasia after wire femoral injury, No atheroprotection	[22,63,72,76]
Selective activation of Membrane ERα	PaPE	Pathway Preferential Estrogens	Acceleration of re-endothelialization following perivascular electric injury, Activation of eNOS activation, Protection against stroke on a mice model of middle cerebral artery occlusion, No atheroprotection	[24,72,77]
Selective activation of Nuclear ERα	E4	Estetrol (Fetal estrogen)	No acceleration of re-endothelialization, Preserved atheroprotection, angiotensin II-induced hypertension and flow-mediated arteriolar remodeling on a model of ligation of 2 mesenteric arteries, Protection against neointimal hyperplasia after wire femoral injury	[22,25,72,78]
Selective modulation of ERα (agonist AF-1)	TAM	Tamoxifen	No acceleration of re-endothelialization after electric carotid injury, Protection against atheroprotection Protection against neointimal hyperplasia after wire femoral injury, Absence of protection against lymphedema	[22,61,79]

NOER: Nuclear only ER; DPM:Disrupting Peptide Mouse; WD:Western diet.

**Table 2 ijms-21-03244-t002:** Summary of lymphatic effects in mouse models of E2 and pharmacological ligands.

Mouse Models	Names	Description	Effect on Vascular Protection	Reference
Inactivation of ERα on blood and lymphatic endothelial cells (Inactivation also in hematopoietic cells)	Tie2-CRE-ERαL2L2	Expression of Cre recombinase under Tie2- promoter	Increased lymphedema Lymphatic leakage, lower VEGFR3 expression on LEC, hyperdilated lymphatic capillaries	[15,22,62,68,73]
Selective modulation of ERα (agonist AF-1)	TAM	Tamoxifen	Absence of protection against lymphedema, no effect on lymphatic capillary density, decrease of VEGFD, Lyve-1 and VEGFR3 expression on LEC, decrease of LEC sprouting and migration.	[15,22,62,68,73]

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
