# Peer review of "The Impact of Estrogen Receptor in Arterial and Lymphatic Vascular Diseases"

_ijms, 2020, doi:10.3390/ijms21093244_

Round 1

Reviewer 1 Report

The lower incidence of CVD in pre-menopausal women compared to men are well-known documented. This protection is attributed to the protective effect of estrogens that exert many beneficial effects against arterial diseases, including vasodilatation, acceleration of healing in response to arterial injury, arterial collateral growth and atheroprotection. More recently, with the visualization of the lymphatic vessels, the impact of estrogens on lymphedema and lymphatic diseases started to be elucidated. These estrogenic effects are mediated not only by the classic nuclear/genomic actions via the specific estrogen receptor (ER) α and β, but also by rapid extra-nuclear membrane initiated steroid signaling (MISS effects). The Estrogen Receptors are expressed by endothelial, lymphatic and smooth muscle cells in the different vessels. In this review, the authors tend to summarize the complex vascular effects of estrogens and selective estrogen receptor modulators (SERMs) that have been described using different transgenic mouse models with selective loss of ERα function and numerous animal models of vascular and lymphatic diseases. Although this work is interesting, this manuscript needs a lot of work.

  1. The rationale of choosing lymphatic vessels together with vascular effect in gender difference is unclear. What is the significance of putting these two together? How much can we attribute to lymphatic vessels in gender difference of CVD? This evidence should be presented and discussed.
  2. English expression needs major improvement. Font varies in certain sections.” some of the numerous effects of estrogens” sounds bad. “Mice models”.
  3. A better description of gender difference in lymphatic and vascular systems should be inserted.
  4. A section of gender difference in cardiovascular function is needed early in the review citing the following papers (PMID. 30943769; 21763763; 20598716; 19214173; 16388094; 32142379).
  5. The table is nice although the mixed presentation of protection and accentuation of vascular function is confusing. The authors should distinguish lymph and vessels here.
  6. Atherosclerosis and ischemia are discussed although other lymphatic vascular disease should also be discussed in term of gender difference. The selection of disease entities is unclear? More clinical aspects are needed.

Author Response

Reviewer

- The rationale of choosing lymphatic vessels together with vascular effect in gender difference is unclear. What is the significance of putting these two together? How much can we attribute to lymphatic vessels in gender difference of CVD? This evidence should be presented and discussed.

Line 753: “While the lymphatic vasculature was first described by Hippocrates (460–377 B.C.), it remains the omitted part of the circulation. Recent studies demonstrated the role of the lymphatic system in cardiovascular diseases including atherosclerosis, hypertensions, heart ischemia and reverse cholesterol transport. Regarding the central role of the lymphatics in the maintenance of fluid homeostasis, it is not surprising to observe sex-related differences associates to the lymphatic function.”

This comment has been added to the discussion

- English expression needs major improvement. Font varies in certain sections.” some of the numerous effects of estrogens” sounds bad. “Mice models”.

Line 738: « some of the numerous effects of estrogens » has been changed for « some of the various effects of estrogens”

- A better description of gender difference in lymphatic and vascular systems should be inserted.

A gender difference paragraph has been added to the manuscript for the vascular system:

Line 144: “1. Importance of estrogens in sex difference”.

Line 531: A paragraph for gender difference in the lymphatic vasculature has been added to the section 5 (ERa and lymphatic vasculature).

As to main scope of the review is to describe the vascular effect of the estrogen receptors, which is express in male and female, we did not focus entirely to the sex difference as this would likely be the subject of a review in itself.

- A section of gender difference in cardiovascular function is needed early in the review citing the following papers (PMID. 30943769; 21763763; 20598716; 19214173; 16388094; 32142379).

Line 144: This section has been added to the manuscript.

- The table is nice although the mixed presentation of protection and accentuation of vascular function is confusing. The authors should distinguish lymph and vessels here.

Table 1 has been divided into 2 tables: Table1 is now representative of the blood vascular effects of the estrogen receptors (ER), Table 2 is representing the lymphatic effects of ER.

- Atherosclerosis and ischemia are discussed although other lymphatic vascular disease should also be discussed in term of gender difference. The selection of disease entities is unclear? More clinical aspects are needed.

Line 531: A paragraph concerning the lymphatic diseases in term of gender difference has been added to the manuscript. More clinical aspects have been associated to this part.

Reviewer 2 Report

The present paper aims to summarize the vascular actions of estrogens on vessels and lymphatics, considering several animal models and clinical studies.

A few minor changes are needed, as follows:

Please explain every abbreviation before using it!

Introduction, page 3, line 106; You mention “mouse model”. You mention also other animal studies and clinical studies. Please mention it in your objectives!

Please include a paragraph emphasizing gender differences in vascular function in different studies (Seeland U, Demuth I, Regitz-Zagrosek V, et al. Sex differences in arterial wave reflection and the role of exogenous and endogenous sex hormones: results of the Berlin Aging Study II. J Hypertens. 2020 Mar 11. doi: 10.1097/HJH.0000000000002386 and Mozos I, Maidana JP, Stoian D, et al. Gender Differences of Arterial Stiffness and Arterial Age in Smokers. Int J Environ Res Public Health. 2017 May 26;14(6). pii: E565. doi: 10.3390/ijerph14060565).

Page 4, line 120: “trogen”?

Table 1. It is difficult to see what is red or green.

A few words of the role of estrogen regulation on miRNA profile in endothelial cells would be appropriate (Vidal-Gómez X, Pérez-Cremades D, Mompeón A, et al. MicroRNA as Crucial Regulators of Gene Expression in Estradiol-Treated Human Endothelial Cells. Cell Physiol Biochem. 2018;45(5):1878-1892. doi: 10.1159/00048791 and Pérez-Cremades D, Mompeón A, Vidal-Gómez X, et al. miRNA as a New Regulatory Mechanism of Estrogen Vascular Action. Int J Mol Sci. 2018 Feb 6;19(2). pii: E473. doi: 10.3390/ijms19020473)

Author Response

- Please explain every abbreviation before using it!

We apologize for this error; abbreviation have been completed.

- Introduction, page 3, line 106; You mention “mouse model”. You mention also other animal studies and clinical studies. Please mention it in your objectives!

Line 142: According to reviewer 2 instructions, the objectives of the review have been emphasized: “This review will focus on the impact of estrogen receptors on arterial and lymphatic vasculature in pathological condition.”

- Please include a paragraph emphasizing gender differences in vascular function in different studies (Seeland U, Demuth I, Regitz-Zagrosek V, et al. Sex differences in arterial wave reflection and the role of exogenous and endogenous sex hormones: results of the Berlin Aging Study II. J Hypertens. 2020 Mar 11. doi: 10.1097/HJH.0000000000002386 and Mozos I, Maidana JP, Stoian D, et al. Gender Differences of Arterial Stiffness and Arterial Age in Smokers. Int J Environ Res Public Health. 2017 May 26;14(6). pii: E565. doi: 10.3390/ijerph14060565).

A gender difference paragraph has been added to the manuscript for the vascular system:

Line 144: “1. Importance of estrogens in sex difference”.

Line 531: A paragraph for gender difference in the lymphatic vasculature has been added to the section 5 (ERa and lymphatic vasculature).

As to main scope of the review is to describe the vascular effect of the estrogen receptors, which is express in male and female, we did not focus entirely to the sex difference as this would likely be the subject of a review in itself.

- Page 4, line 120: “trogen”?

We apologize for this error; the world has been corrected.

- Table 1. It is difficult to see what is red or green.

Table 1 has been divided into 2 tables: Table1 is now representative of the blood vascular effects of the estrogen receptors (ER), Table 2 is representing the lymphatic effects of ER.

- A few words of the role of estrogen regulation on miRNA profile in endothelial cells would be appropriate (Vidal-Gómez X, Pérez-Cremades D, Mompeón A, et al. MicroRNA as Crucial Regulators of Gene Expression in Estradiol-Treated Human Endothelial Cells. Cell Physiol Biochem. 2018;45(5):1878-1892. doi: 10.1159/00048791 and Pérez-Cremades D, Mompeón A, Vidal-Gómez X, et al. miRNA as a New Regulatory Mechanism of Estrogen Vascular Action. Int J Mol Sci. 2018 Feb 6;19(2). pii: E473. doi: 10.3390/ijms19020473).

A short paragraph concerning the estrogen regulation on miRNA profile in endothelial cells has been added to the manuscript:

Line 495: “More recently, growing evidence has established regulation of numerous microRNA (miRNA) involved in the regulation of endothelial function by E2 through ER, providing new insights by which estrogen can modulate vascular action ((Vidal-Gómez X, 2018). This study validated the presence of ERE in the regulatory region of endothelial miRNA, showing E2-induced increase of miR-30b-5p, miR-487a-5p, miR-4710, miR-501-3p and decreased of miR-378h and miR-1244. miRNA may therefore act as crucial epigenetic regulators of gene expression induced by E2 in  endothelial cells and could contribute to further characterize the role of specific miRNA in vascular actions mediated by E2 in physiological and pathological conditions ((Daniel Pérez-Cremades 2018).”
